# Equity in Access: A Mixed Methods Exploration of the National Disability Insurance Scheme Access Program for the Kimberley Region, Western Australia

**DOI:** 10.3390/ijerph18178907

**Published:** 2021-08-24

**Authors:** Caitlyn S. White, Erica Spry, Emma Griffiths, Emma Carlin

**Affiliations:** 1Kimberley Aboriginal Medical Services, Broome 6725, Australia; phr@kamsc.org.au (C.S.W.); research@kamsc.org.au (E.S.); emma.griffiths@rcswa.edu.au (E.G.); 2The Rural Clinical School of Western Australia, University of Western Australia, Broome 6725, Australia

**Keywords:** National Disability Insurance Scheme, Aboriginal, disability, remote community connector, Aboriginal Community Controlled Health Service

## Abstract

This study explored the process and early outcomes of work undertaken by a program to increase Aboriginal people’s awareness of, and access to, the National Disability Insurance Scheme (NDIS). This ‘Access Program’ was implemented through the Aboriginal Community Controlled Sector in the remote Kimberley region of Western Australia. Access Program staff were interviewed to explore the strengths, challenges, and future directions of the program. The demographics, primary disability types, and NDIS access outcomes for clients who engaged with the program in the first 12 months of its implementation have been described. The Access Program engaged with 373 clients during the study period and assisted 118 of these to achieve access to the NDIS. The program was reported as successful by staff in its aim of connecting eligible people with the NDIS. Vital to this success was program implementation by the Aboriginal Community Controlled Sector. Staff in these organisations held community trust, provided culturally appropriate services, and utilised strengths-based approaches to overcome barriers that have historically hindered Aboriginal people’s engagement with disability services. Our results demonstrate the Access Program is a successful start in increasing awareness of, and access to, the NDIS for Aboriginal people in the Kimberley region. Much work remains to assist the large number of Aboriginal people in the Kimberley region believed to be eligible for NDIS support who are yet to achieve access.

## 1. Introduction

In Australia, Aboriginal and Torres Strait Islander people experience disability at twice the rate of other Australians [1]. Historically, Aboriginal and Torres Strait Islander people’s access to and involvement with disability support services has been constrained [2]. This suboptimal engagement was related to the underfunding of the disability sector, complexity in accessing disability supports, and fragmented provision of services [3,4]. Each Australian state had different systems for allocating funding [3] that were subject to government budget cycles with funding often allocated to service providers rather than individuals [4]. This resulted in significant unmet demand for disability services with access to services sometimes dependent on how the disability was acquired or where a person lived [4]. While many of these challenges were also experienced by non-Aboriginal people with disabilities, these barriers were amplified for Aboriginal and Torres Strait Islander people [5,6].

Launched in 2013 and progressively rolled out since 2016, the National Disability Insurance Scheme (NDIS) was developed to provide a better funding model for all Australians with permanent and significant disabilities [7]. The NDIS is a national system for allocating funding for disability support services to individuals based on their needs [8]. The NDIS has been described as a ‘once in a generation reform’ [9] and has been welcomed by Aboriginal peak bodies such as the First Peoples Disability Network as an opportunity to improve engagement and service provision for Aboriginal and Torres Strait Islander people [2]. Notwithstanding the general support for the NDIS, Aboriginal disability peaks and advocates have highlighted several potential issues regarding its approach to implementation.

The NDIS restructures funding from services to individuals promoting greater ‘choice and control’ [10]. However, for remote communities (characterised by small populations and vast distances from service centres) there is concern that demand may be insufficient to secure the local supply of services needed [11]. Second, the NDIS application process can be complex and often inaccessible for Aboriginal and Torres Strait Islander people with a disability [11,12]. The First Peoples Disability Network and the National Aboriginal Community Controlled Health Organisation have advocated for culturally appropriate engagement strategies in the implementation of the NDIS to ensure equitable access to the scheme for Aboriginal and Torres Strait Islander people [2,11,13]. Key to this advocacy was the recommendation that the Aboriginal Community Controlled Health Sector is essential for the successful delivery of the NDIS to Aboriginal and Torres Strait Islander people with a disability [13]. Third, advocates have raised the need for accessible support and training for people to self-manage their NDIS plans once access to the scheme is granted, especially for those who may experience language or cultural barriers [14]. It has been recognised that for some Aboriginal people access to a client advocate to support the participant in navigating complex disability systems is necessary [2].

The governing body of the NDIS, the National Disability Insurance Agency (NDIA), has recognised the different needs of certain groups in accessing the NDIS [11,15]. The NDIA Rural and Remote Strategy [16] and Aboriginal and Torres Strait Islander Engagement Strategy [17] outline how the NDIA aims to engage with Aboriginal and Torres Strait Islander people in culturally appropriate ways [16,17]. These documents provide valuable foundations and demonstrate that the NDIA is cognisant that Aboriginal and Torres Strait Islander access in remote and regional Australia demands innovation, investment, and partnerships [18,19,20]. They do not, however, provide information to guide the implementation of the NDIS at a practical level. As overarching documents, they lack the granular detail of recent Aboriginal-led research which identifies the approach to engagement (strengths-based) and the characteristics of the service provider (Aboriginal Community Controlled) as key pillars in NDIS engagement and service delivery [21]. Similarly, the NDIA, to date, has not provided a framework to help build the capacity of non-Aboriginal community controlled service providers to achieve positive engagement with Aboriginal and Torres Strait Islander individuals, families, and services [22].

The Kimberley NDIS “Access Program” was developed after a $4.6 million investment by the NDIA into Aboriginal Community Controlled Health Services in regional and remote Western Australia to improve awareness of, and access to, the NDIS for Aboriginal people [23]. This Access Program funds the work of two program roles: Remote Community Connectors (RCCs) and Evidence, Access and Coordination of Planning coordinators (EACPs). The RCC role operates in other parts of remote Australia as part of the NDIA’s Community Connector Program for groups requiring additional support to access the NDIS [24]. The EACP role was a newly funded initiative and currently exists only in Western Australia. This was advocated for by the Aboriginal Health Council of Western Australia in response to the need for two distinct types of support in accessing the NDIS for remote Aboriginal people with a disability. First, support in raising awareness and understanding of the NDIS through the RCCs; second, in providing support to gather evidence and complete the application process within the timeframes set out by the NDIA.

The Kimberley Access Program is nested within the NDIA funded Kimberley Supports programs managed by Kimberley Aboriginal Medical Services (KAMS) and delivered by a consortium of regional Aboriginal Community Controlled Organisations and Aboriginal Community Controlled Health Services.

This research aims to explore the work being undertaken by Kimberley Access Program staff who are tasked with identifying and connecting eligible Aboriginal community members to the NDIS. We aim to contribute to an understanding of the challenges, successes and future directions of ensuring equitable access to the NDIS for Aboriginal people in the Kimberley.

## 2. Materials and Methods

### 2.1. Research Design

A mixed methods approach [25] was used to explore the process and early outcomes of the NDIS Access Program for Kimberley Aboriginal people and organisations. A qualitative descriptive approach [26] was used to engage Access Program staff respondents around the following four topics: engagement with potential NDIS participants; facilitating next steps with potential NDIS participants; experience of the program’s outcomes; and barriers to and enablers of the program. A quantitative component descriptively analysed participant engagement data collected prospectively by Access Program staff during the early implementation phase of the project (31 October 2019 to 16 November 2020). This research is aligned to the National Health and Medical Research Council guidelines on ethical research with Aboriginal and Torres Strait Islander people [27].

### 2.2. Research Priorities and Ethics

This research was undertaken at the request of KAMS. KAMS, established in 1986, is the peak Aboriginal Community Controlled Health Organisation in the Kimberley region and is committed to progressing an evidence base for its emergent disability activities. All consortium organisations involved in the Access Program were consulted in the development of the research proposal and provided written support for the project. Consortium organisations were provided with a copy of the manuscript to review prior to submission of the article for publication. This process of feedback ensured the publication is reflective of the experiences and perceptions of the organisations within the consortium.

The project was endorsed by the Kimberley Aboriginal Health Planning Forum Research Subcommittee prior to receiving formal ethics approval from the Western Australian Aboriginal Health Ethics Committee (Project 970). Both of these committees include Aboriginal leadership and exist to support and promote research that reflects and is responsive to the needs of Aboriginal people [12,28].

### 2.3. Setting

#### 2.3.1. The Kimberley Region

The Kimberley region (see Figure 1), in the northwest of Western Australia, spans more than 400,000 square kilometres and is one of the most sparsely populated regions of Australia with a population density of 0.09 per square kilometre [29]. The recorded resident population is approximately 34,000 people, with 42% of the population identifying as Aboriginal [30]. The population lives in remote communities, several small towns, and one medium-sized town, which functions as the regional centre. Health care is mainly provided by the Aboriginal Community Controlled Health Services and State Government health services. Aboriginal people in the Kimberley region continue to experience the intergenerational impacts of colonisation on health and wellbeing [18]. These impacts include experiencing high levels of socioeconomic disadvantage, an excessive burden of chronic disease, lower life expectancies than non-Aboriginal people, and a high burden of disability [1,31].

#### 2.3.2. The Access Program

RCCs and EACPs are employed by eight consortium organisations based in four towns in the Kimberley region. RCCs and EACPs have provided support to potential NDIS participants in at least 12 remote communities and six Kimberley towns. The RCC role includes building community-level awareness and understanding of the NDIS and engaging with individuals and families who may be eligible for the NDIS. RCCs then support clients through the NDIS access process. RCCs refer to EACPs who assist clients to apply for the NDIS and access NDIS funded services.

Once informed consent has been obtained, the EACP gathers the necessary evidence from the client to complete an NDIS access request form. EACPs may access the medical record with patient consent to find appropriate supporting evidence for their client’s application. When medical records are held in non-consortium clinics or historical records are required, EACPs may submit applications to other services for release of information. If appropriate evidence for an application is not available in the medical record, EACPs may organise functional assessments for their clients with an allied health professional or general practitioner. The process of gathering evidence can be time consuming and labour intensive. The EACP submits the completed form and supporting documentation to the NDIA’s National Access Team for remote areas in Western Australia on behalf of their client. Clients may consent to EACPs acting as their delegate to coordinate communication between themselves and the NDIA throughout this process. If a client does not qualify for access to the NDIS after their access request form has been reviewed by the NDIA (termed “access not met”), the EACP may assist them to appeal the decision or link the client with other support services as appropriate. If the client does meet access to the NDIS on review by the NDIA (termed “access met”), the EACP will assist the participant to prepare for a planning meeting with the NDIA (termed “pre-planning”). RCCs and/or EACPs may assist with coordinating planning meetings and may attend with the participant as per their wishes. EACPs and RCCs may be employed by different Aboriginal Community Controlled Organisations and Aboriginal Community Controlled Health Services but work in partnership to support both individual and community level outcomes.

### 2.4. Participants

A purposive sampling frame of the Access Program staff was used for the qualitative component of the study. Twenty Access Program staff were employed at the time of data collection (October 2020 to January 2021) and were invited to participate. In addition to Access Program staff, managers and executives were invited to participate to gain additional background and perspectives on the Access Program.

Quantitative analysis included data from a total of 373 potential NDIS participants who had engaged with the Access Program between 31 October 2019 and 16 November 2020.

### 2.5. Data Collection and Analysis

#### 2.5.1. Quantitative Data Collection and Analysis

Quantitative analysis utilised a database provided by the Regional Access Program Coordinator. This data set consisted of de-identified information from all participants referred to an EACP from 31 October 2019 to 16 November 2020. Variables within the data set included age, community of residence, source of referral, primary disability, secondary disability, and stage of NDIS application. The primary disability listed was categorised to protect confidentiality using an impairment of functioning approach [32]. The categories of intellectual, physical, psychiatric, neurological (including acquired brain injury), and sensory (including vision, hearing, and speech) disability were used, with the addition of a category that included both Developmental Delay and Fetal Alcohol Spectrum Disorder (FASD), as these disabilities accounted for a substantial number of EACP referrals, particularly in children, and can affect multiple functional domains [33,34]. Location of residence was classified according to the Modified Monash Model [35].

Referrals to the EACP were summarised by referral source, outcome of referral, category of disability, and time to referral outcome. Descriptive analysis was completed in Stata Statistical Software: Release 16 (StataCorp, College Station, TX, USA, 2019).

#### 2.5.2. Qualitative Data Collection and Analysis

In total, 14 semi-structured interviews were conducted with 11 Access Program staff and three program managers/executives. Of these interviews, eight were conducted face to face (including two group interviews with five and two participants) while six were conducted over video conference. Interviews were conducted during working hours at staff members’ places of work. All interviews were undertaken by a female non-Aboriginal clinical researcher (CW) and voice recorded with written participant consent.

Interviews were professionally transcribed into individual Microsoft Word documents. These were imported into NVivo 12 (QSR International, Burlington, USA) and then coded and analysed using a directed qualitative content analysis approach [36]. All interviews were coded into categories by EC and CW, resulting in four nodes: Access Program Roles, Access Program Enablers, Access Program Challenges, and Access Program Workforce. Nodes were reviewed by the research team (EC, ES, CW) and analysed to generate preliminary themes (perceptions of disability, engagement as a precursor to NDIS access, and importance of ‘getting it right’). These themes were reviewed with the research team (EC, ES, CW and EG) and key quotes chosen. Analysis was also presented to key staff from the Access Program prior to finalisation. The research team included and prioritised Aboriginal voices in the coding and analysis, and research team member ES provided cultural oversight of the use of these quotes.

## 3. Results

### 3.1. Quantitative Results

Between 31 October 2019 and 16 November 2020 a total of 373 people were referred to an EACP in the Kimberley region. Characteristics of these clients are outlined in Table 1. Fifty-five percent of clients (206/373) had completed the referral process. Of these, 118 clients (118/206, 57%) had access met either by being approved for NDIS access (*n* = 102) or being confirmed as a pre-existing NDIS participant (*n* = 16). Eighty-eight clients (24%) did not access the NDIS because of not meeting access requirements (*n* = 72) or self-withdrawing from the referral process (*n* = 16).

The remainder (167/373, 45%) had referrals in progress. Of these, 66 (66/167, 40%) were in early engagement with Access Program staff, 76 (76/167, 46%) were having evidence gathered, 23 (23/167, 14%) had completed their Access Request Form, and the referral status for two (2/167, 1%) clients was unknown.

Intellectual (56/373, 15%), physical (50/373, 13%), and psychiatric disability (50/373, 13%) categories were the most common primary disability type recorded for participants referred to an EACP (Table 2).

Time from EACP referral to formal submission for NDIS access was highly variable, ranging from 1 to 265 days. Similarly, time from formal referral to access approval ranged from 1 to 83 days.

### 3.2. Characteristics of the Qualitative Study Sample

Of the 20 Access Program staff eligible to participate, a total of 11 program staff were interviewed. Nine of the interviewees were employed in access program delivery (RCC or EACP) and two were access program coordinators. The remaining nine Access Program staff did not respond to an invitation to be involved or were not available when the interview took place. An additional three interviews were held with consortium executives or managers. Out of the 14 interviews conducted, four staff members interviewed were Aboriginal and three were male. All staff who participated in the interviews provided written informed consent.

### 3.3. Qualitative Analysis

#### 3.3.1. Theme 1: Perceptions of Disability

When reflecting on the Access Program, staff discussed Aboriginal people’s perceptions and experiences of disability. Many staff reported that disability is often ‘not recognised’ by the individual or by the individual’s family group. High levels of resiliency and normalisation of the disability were raised as contributing factors:
*“I think one of the hardest parts of it* [engaging potential clients] *is, people not realising themselves that they have a disability…people are just so resilient in communities and they just get on with things… So, when you go and speak to people…‘No, we’re right’. They think they’re not disabled, that it’s normal.”*Remote Community Connector

Additionally, Access Program staff discussed the fear and stigma that Aboriginal community members associated with concepts of disability:
*“…a lot of people have got that stigma ‘well I don’t want my child being labelled as disabled’ and we’re explaining to them it’s not being labelled as ‘disabled’ it’s just getting the support that they may need at school to help them learn and help them. So trying to explain that to a lot of the community so that they don’t have that stigma …the shame factor…”*Evidence, Access and Coordination of Planning Coordinator

Other staff discussed how disability often had a ‘different meaning’ within Aboriginal communities and was often not seen as central to a person’s identity:
*“It’s a sensitive issue disability. You’ve got a lot of people who have had permanent disabilities for a long time or born with* [a disability]. *It doesn’t change their standing in community or their decision making or where they sit within their family.”*Kimberley Consortium Executive/Manager

Staff discussed how it was important to be aware of these differing perceptions of disability and not to approach disability as a deficit or something to be ‘fixed’ with potential clients and their families. Staff reflected on the need for conversations to be enquiry based, have a focus on recognising the strength and ‘resilience’ of an individual and family, and include key family members.

#### 3.3.2. Theme 2: Engagement—A Precursor for NDIS Access

Many staff reflected that historically the Kimberley region has experienced a lack of available or suitable disability support services. This was perceived to complicate people’s engagement with the NDIS as trust had been broken and people were reticent to ‘waste their time’ with another service. Engagement at the individual and community level was perceived as essential to work through barriers to accessing disability services and promoting awareness of the NDIS. RCCs and EACPs described engagement as requiring high levels of flexibility, commitment, and support:
*“The beauty of your RCCs, their skill set isn’t necessarily sitting in front of a computer and gathering evidence…They’ve got the ability to actually engage people around NDIS.”*Kimberley Consortium Executive/Manager
*“I run around, try and find the clients, go to this house and that house, finally find them, and then I’ll do meetings under a tree, sitting there having a yarn just like it’s nothing else.”*Remote Community Connector

Access staff described their knowledge of the community as helping them to make targeted visits to certain families:
*“Well a lot of it is just knowledge of the community… And knowledge of actual people. You know like maybe potential participants because you know them in a social* [setting]… *and working here at* [name of service] *you see people coming in and coming out and you know they may have a disability”.*Evidence, Access and Coordination of Planning Coordinator

Staff discussed the positive impacts of having Aboriginal Community Controlled Organisations deliver the Access Program. These organisations were described as trusted, culturally appropriate, and holding pre-existing relationships with clients. This in turn enhanced the acceptability of the Access Program staff working to raise awareness and facilitate access to the NDIS:
*“They know the name* [Organisation Name], *they know the brand already and they know we’ll be there… We don’t go away. So, we’re in.”*Kimberley Consortium Executive/Manager

Access Program staff described the engagement with clients as enduring with many EACPs continuing to work with clients after access had been met. This was to ensure clients had appropriate support and advocacy when engaging with NDIA staff to develop their support plan:
*“During this time, you liaise or become the person that connects the NDIA planners to the participant and help with organising the meetings and setting that up. Making sure that it’s conducted in an appropriate manner, make sure everyone’s or the person is comfortable and is best prepared as it can be. Quite often I sit in on these meetings to support and also go through, basically prompt the person if there’s something that’s been missed.”*Evidence, Access and Coordination of Planning Coordinator

It was identified that Access Program staff also met with client frustration when timeframes for access decisions from the NDIA were lengthy, access to the NDIS was denied, or the appropriate disability support services were unavailable in communities.

#### 3.3.3. Theme 3: The Importance of ‘Getting It Right’

Staff reflected on how the program had provided a powerful advocacy service for the region that had resulted in a shift in the way that NDIA reviewed Kimberley NDIS applications. Access Program staff reported instances when the NDIA were responsive to issues affecting clients’ access to the NDIS. EACPs discussed being an important source of contact for the NDIA to clarify issues in applications in order to facilitate meeting access and avoid lengthy decision-making delays for clients:
*“…so if there’s a birth certificate missing or the dates don’t match Centrelink and different things like that, instead of rejecting it and sending a letter straight to the participant, they’ll talk to the Evidence and Access person who’s on the third-party consent form.”*Access Program Coordinator

Despite the responsiveness of the NDIA with application review, Access Program staff still reported that the access pathway to the NDIS could be challenging for clients due to bureaucratic inflexibility. For example, there could be pressure from the NDIA to quickly move a client to the planning stage after access being met despite a client not yet being ready to proceed. Locally based Access Program staff were well positioned to advocate for clients to proceed at their own pace:
*“…it’s about being that person to liaise with the NDIA to say ‘Hey, this person might not be ready or they might not be in community, or they haven’t done their pre-planning’… we had someone with acute psychosis in the mental health here and they still thought it was appropriate to have a planning meeting while they were there and not in a healthy state of mind in a familiar environment… They’re* [the NDIS] *very keen to get things happening really soon. A bit too soon in some cases.”*Evidence, Access and Coordination of Planning Coordinator

Access Program staff commonly reported that NDIA resources were not culturally appropriate for Aboriginal people in the Kimberley. ‘Easy read’ versions of booklets provided by the NDIA did not meet the cultural needs of remote Aboriginal people, and Access Program staff created their own resources as a result of this gap. Access Program staff and managers reflected that NDIA training was not targeted to the unique needs of the Kimberley Access Program workforce, and the onus on developing responsive and appropriate training became an issue for the consortium to resolve.

Staff noted that confusion existed amongst communities regarding the eligibility requirements for the NDIS. In particular, a common perception was that those on a government-issue disability support pension would be automatically eligible for the NDIS. Access staff noted they were required to explain that while both supports came from the federal government and included the word ‘disability’ the eligibility criteria are very different.

Several staff reflected on the diversity of the program’s workforce with many coming from outside the health and disability sector. While the diversity was seen as a strength of the program, it highlighted the need for ongoing training and a baseline knowledge in disability:
*“Sometimes I’m finding that it’s* [the referral is for] *health reasons but…there’s not a disability, so we’ve got to get our people to understand the difference as well about what a disability is.”*Evidence, Access and Coordination of Planning Coordinator

Overall staff felt positive about the program and the regional efforts that had been made in terms of training, program set up, and clients meeting access. Program managers noted the value of the large consortium of community controlled organisations delivering the program. Specifically, partners brought value to the table in terms of local knowledge and the ability to implement place-based engagement responses across the vast and remote Kimberley region. The program was frequently described as ‘successful’ by staff. Despite this, there was a strong sense that the work had just begun with staff voicing fears that the project would not be funded after the current contract ends in the middle of 2021:
*“There is maybe a lot more people in the Kimberley who would be eligible for NDIS. I think there’s probably still a lot more work to be done in the way disability is described and talked about, just generally, so that community members awareness starts to lift…”*Kimberley Consortium Executive/Manager

## 4. Discussion

This research has demonstrated the importance of the Kimberley Access Program as described by a review of audit data and consultation with program providers. The program positively contributes to evidence that place-based approaches to engagement are successful in promoting awareness of and access to the NDIS for a population that experiences profound social inequality and exclusion [37]. Congruent with other studies, our research suggests Aboriginal people with a disability often participate in community life regardless of their disability and often do not identify as having a disability [2,38,39]. Normalisation, fear of stigmatization, and a history of culturally inappropriate services further complicate Aboriginal people’s self-identification with, and perceptions of, disability [7,38,40,41]. For these reasons, Aboriginal perceptions of disability have been identified as a barrier in engagement with disability services [6,8,41]. Access Program staff displayed understanding of these perceptions and described ways in which they were able to help overcome the barriers. This included a strengths-based [39] and family systems approach [42] to identifying supports that could build on the strengths and resilience of the individual rather than adopting a deficit-based approach to support [43].

Access Program staff and managers discussed the importance of Aboriginal Community Controlled Organisations leading engagement. Aboriginal-led approaches were identified as best placed to overcome the historical and systemic factors that contribute to low levels of engagement by Aboriginal people with a disability [2,13,43]. The personal qualities Access Program staff described as important to their role, including flexibility, commitment to outcomes and knowledge of the community can all be identified as expressions of culturally secure engagement with Aboriginal people [6]. Previous research has demonstrated the disconnect between the NDIA’s understanding of what is involved in engagement work compared to what engagement looks like to local community connectors [41]. A non-Aboriginal Community Controlled Health Service led approach to NDIS implementation with Aboriginal and Torres Strait Islander communities in the Northern Territory and Queensland found that attempts from the NDIA to engage with communities had been perceived as cursory, inappropriate, and ineffective by the communities they sought to engage with [41]. In contrast, an Aboriginal Community Controlled Health Service led approach in remote Central Australia found that a model where Aboriginal coworkers worked alongside disability workers was effective in delivering culturally safe and acceptable services to Aboriginal people with a disability [18,19]. Our study contributes to an evidence base that demonstrates the importance of Aboriginal Community Controlled responses in achieving equitable access for remote Aboriginal people with a disability.

While the proportion of those with a physical disability who engaged with the Access Program was lower than observed in the national dataset, the proportion of those with non-physical disabilities (intellectual, psychiatric) was higher [44]. It has been noted that people with psychosocial disabilities face particular barriers to NDIS access [45]. For these individuals, it may be especially challenging to obtain the documentation required by NDIA to prove permanent and significant disability [45]. The Access Program provides support to overcome these barriers with EACPs assisting clients to obtain critical evidence for their NDIS application. We found key EACP activities included accessing the client’s medical record with their consent, requesting health records from other services, and/or arranging functional assessments. Access Program staff were able to use their existing community knowledge and connections to identify people with psychosocial disabilities that may not have otherwise been identified as eligible and begin the engagement process.

With relatively low rates of self-referrals to the Access Program, referral data indicates that engagement has occurred largely from RCCs, again, using their local knowledge. Another important source of referral is through clients engaging with other local organisations including the health sector, child and family services, and the justice services. The Access Program has demonstrated a promising brokerage role between these organisations and the NDIS.

Despite the promising success of the Kimberley Access program, evidence suggests that for Aboriginal people in rural and remote Australia more work needs to be done [16]. The NDIA report that as of 30 September 2020 there were 1069 active NDIS participants (Aboriginal and non-Aboriginal) in the Kimberley and Pilbara regions of Western Australia combined [46]. Estimated rates of disability amongst Aboriginal and Torres Strait Islander people, the demographic characteristics of the Kimberley region, and findings from this paper suggest many more eligible Aboriginal people are yet to achieve access to the NDIS [30,47].

### Strengths and Limitations

This study is the first of its kind for the Kimberley region. It provides a regional level understanding of the NDIA funded Access Program and a profile of the individuals who are engaging with the program. It is the first study to explore the novel role of the EACP, which is unique to Western Australia. We note as a limitation that the qualitative component of this study did not include any potential or active NDIS participants that had engaged with the Access Program. Missing data in the quantitative analysis limited the conclusions that could be drawn from its analysis. It is recommended that future research engage with potential and active NDIS participants to explore their NDIS journey from access through to the provision of support they receive via their NDIS funded plan.

## 5. Conclusions

As a ‘once in a generation’ reform, the NDIS presents the opportunity for Australians living with a disability to access disability support services to reach their goals with greater choice and control. For Aboriginal and Torres Strait Islander people living in remote Australia, access to and engagement with disability services has historically been limited. Successfully implementing the NDIS with this priority population, who are disproportionately burdened with disability, is in the interest of both the NDIA and the Aboriginal Community Control Sector. The Kimberley Access program has demonstrated success in engaging remote Aboriginal clients who are eligible for the NDIS. Overall, this study demonstrates how the two parts of the Access Program work both together, and with other community services, to support clients in accessing the NDIS.

## Figures and Tables

**Figure 1 ijerph-18-08907-f001:**
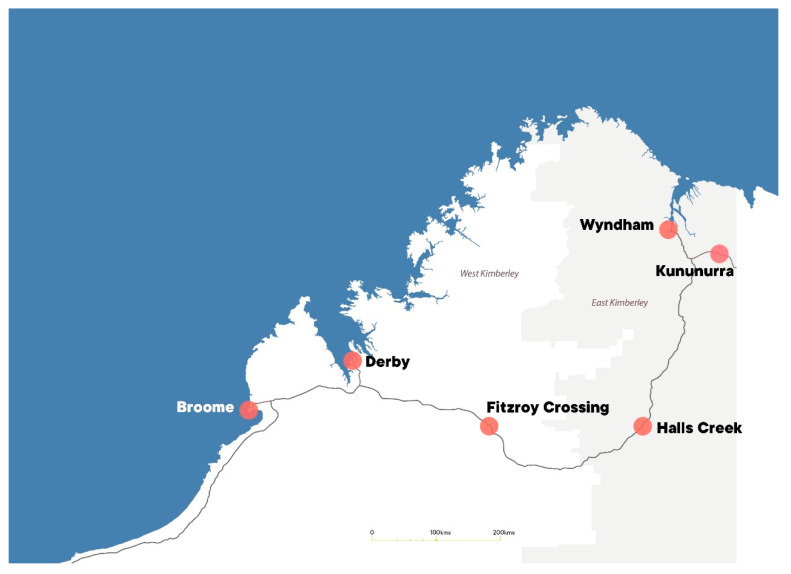
Map of the Kimberley Region, Western Australia.

**Table 1 ijerph-18-08907-t001:** Characteristics of those referred to an EACP in the Kimberley region 31 October 2019 to 16 November 2020.

	Number of Clients	Percentage of All Referrals
**Age**		
0–9 years	53	14%
10–19 years	68	18%
20–29 years	35	10%
30–39 years	31	8%
40–49 years	43	12%
50–64 years	68	18%
65 years and over	7	2%
Unknown/missing data	68	18%
**Remoteness**		
Remote	155	41%
Very remote	211	57%
Unknown/missing data	7	2%
**Referral Source**		
Remote community connector	151	41%
Aboriginal Community Controlled Health Service	89	24%
Other health service	31	8%
Other organisation	31	8%
National Disability Insurance Scheme	22	6%
Self-referral	15	4%
Unknown/missing data	34	9%
**Total referred**	373	100%

**Table 2 ijerph-18-08907-t002:** Referrals to EACP by primary disability category and access status {*n* (%)}.

	Referral Complete		Referral in Progress
	Access Met	Access Not Met	
Neurological/Acquired Brain Injury	7 (6%)	5 (6%)	5 (3%)
Psychiatric	18 (15%)	7 (8%)	25 (15%)
Physical	18 (15%)	5 (6%)	27 (16%)
Sensory	14 (12%)	2 (2%)	7 (4%)
Intellectual	24 (21%)	5 (6%)	27 (16%)
Developmental delay or FASD	19 (16%)	11 (12%)	18 (11%)
No disability	0 (0%)	26 (29%)	8 (5%)
Unknown/missing data	18 (15%)	27 (31%)	50 (30%)
Total	118 (100%)	88 (100%)	167 (100%)

## Data Availability

Data are available upon request to the authors and conditional on ethics approval.

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
