# Peer review of "Equity in Access: A Mixed Methods Exploration of the National Disability Insurance Scheme Access Program for the Kimberley Region, Western Australia"

_ijerph, 2021, doi:10.3390/ijerph18178907_

Round 1

Reviewer 1 Report

As Australia's lead scholar in Indigenous disability research, I loved reading this paper. I only have a few suggestions:

  1. more content for methodology: please outline how this methodology reflects or progresses already used Aboriginal disability research frameworks and standpoints. See Gilroy, Soldatic, Dew, and Avery
  2. add some content to the background on what Aboriginal research and remote/regional research is uncovering in this space. This has been touched on a little. You could reduce the content on the NDIS background to make room. You could add a paper on Ability Links program published by Gibson. Also Gilroy did some stuff in Brain INjury in remote QLD and also with Anangu people. I note these are raised in the Discussion, very well BTW. I do think the intro reads more like a government report rather than a scholarly paper. 

Reviewer 2 Report

Introduction: 

I think you did a good job in presenting the information needed to understand the background to the issues. It is generally a concise and well written summary, though it does remain challenging to read given the detail and the acronyms. The information about the Remote Community Connector in paragraph 4 and 5 was unclear to me here with para 4 not clearly linked to para 5. I didn’t understand the connectors until more fully discussed in the methods. Introducing a figure or text box to present some of this information may aid readability (for example combining the information in the introduction with that in 2.3.2 line 139).

Methods:

Your description of your qualitative methods needs clarification. Two different references and methods are used to describe the qualitative approach and analysis (qualitative description ref 23 and content analysis 29). The term thematic analysis is also used with a statement about emerging themes, where perhaps it would be more appropriate to say the themes were generated rather than using a word which suggests grounded theory and inductive approaches, particularly as the codes did appear deductive.

The detail in lines 188-196 describing your node categories and then your themes decreased clarity for me, as I queried how the themes have been derived from the nodes, and how your analysis could be inductive after deductive initial coding categories. Removing detail may assist, if in fact the coding categories were not used in your final analysis, and it is not usual to have the themes in the methods.

Line 207-210 – Are you differentiating intellectual disability relate to developmental delay and to other reasons in this category? Are Developmental Delay and FASD one category or two?

The narrative would be improved if your methods and results were ordered by quantitative and qualitative components in the same way.

Results:

I think this would be improved with some removal of detail. For example, line 234-241 repeats some of the information in Table 1. Table 3 provides unnecessary detail, I am not sure if there is a supplementary materials option, otherwise a simple summary sentence would be better. There are too many illustrative quotes, which are also very long, leading to loss of the narrative in theme 2.

Table 2 – I am not sure what ‘access met’ and ‘access not met’ means – is this eligibility?

Discussion:

I felt the strong opening sentences here were not entirely justifiable based on your data, I suggest softening the language away from a statement that your study demonstrates the success of the program. Having no data from consumers of your program is a limitation that comes strongly into play here. The data certainly presented the complexity of the field, insights into the program and the view of the people working within the program that it was important and what strategies were essential.

I think the discussion should decrease its advocacy focus to some degree so as to increase the prominence of what other readers can learn from the program, and potentially apply in their own contexts. For example, lines 453-463 were heartening but I don’t think belonged in this research paper.

Round 2

Reviewer 2 Report

Thank you for considering my review. I feel the manuscript is improved and am happy to recommend it be accepted.